# The NXDC-MEN-301 Study on 5-ALA for Meningiomas Surgery: An Innovative Study Design for the Assessing the Benefit of Intra-Operative Fluorescence Imaging

**DOI:** 10.3390/brainsci12081044

**Published:** 2022-08-06

**Authors:** Walter Stummer, Markus Holling, Bernard R. Bendok, Michael A. Vogelbaum, Ashley Cox, Sara L. Renfrow, Georg Widhalm, Alan Ezrin, Salvatore DeSena, Murray L. Sackman, Joseph W. Wyse

**Affiliations:** 1Department of Neurosurgery, University Hospital Münster, 48161 Münster, Germany; 2Department of Neurosurgery, Mayo Clinic, Phoenix, AZ 85054, USA; 3Department of NeuroOncology, Moffit Cancer Center, Tampa, FL 33612, USA; 4NX Development Corporation, Lexington, KY 40503, USA; 5Department of Neurosurgery, Medical University of Vienna, 1090 Vienna, Austria

**Keywords:** fluorescence imaging study, neurooncological surgery, 5-ALA, meningioma, fluorescence-guided resection, clinical trials

## Abstract

Background: 5-aminolevulinic acid (5-ALA; Gleolan^TM^, NX Development Corps., Lexington, USA) is approved for fluorescence-guided resections of suspected malignant gliomas. Experience has demonstrated that meningiomas also show fluorescence, which may be a useful surgical adjunct. We present an innovative design for a multi-center, prospective study to determine the clinical safety and potential benefit of fluorescence-guided resection of meningiomas with utmost bias reduction. Methods: All patients with suspected meningioma (all grades) receive Gleolan^TM^ 20 mg/kg 2–4 h prior to surgery supported by fluorescence excitation from a blue light source (Blue400, Zeiss Meditech, Oberkochen, Germany; FL400, Leica Microsystems, Heerbrugg, Switzerland). Surgeons are asked whether a residual tumor can be observed to fluoresce under blue light (BL) after the tumor is no longer recognizable using conventional illumination at the end of surgery. In addition, when faced with tissues of uncertain tissue type (so-called “indeterminate” tissue), this study records how often surgeons make a correct decision based on fluorescence and how this influences surgical strategy. The primary endpoint is the percentage of patients in whom one of these two benefits are observed. Other endpoints include the diagnostic accuracy of fluorescence compared to white light (WL) versus correlative histology. For bias reduction, pertinent data are derived from surgical videos reviewed by independent reviewers blinded to surgeons’ assessments of tissue type and fluorescence status. Data will be included from approximately 100 study participants completing the study at approximately 15 centers in the United States, Germany, and Austria. Results: As of May 2022, 88 patients have completed the study. No adverse safety signal has been detected. Conclusions: Preliminary data confirm the feasibility of our study design. Accrual is targeted for completion in the third quarter of 2022.

## 1. Introduction

Fluorescence-guided resection (FGR) using 5-ALA is well established for surgery of malignant gliomas [1,2]. Not only has the value of FGR been tested prospectively for finding residual tumor [3] and improving the extent of resection, but 5-ALA-induced tumor fluorescence also provides help in defining surgical strategy, since tissue fluorescence is usually interrogated repeatedly during surgery, e.g., for initially finding subcortical tumors and for defining the surgical route along tumor margins.

The question has frequently been raised regarding which other brain tumors might profit from fluorescence-guided tumor detection, and it is known in the literature that different brain tumors show various degrees of fluorescence [4]. Among these, meningiomas are potential candidates for fluorescence-guided resection, as they have been shown to demonstrate fluorescence in almost all cases [5,6,7,8,9,10]. At first glance, meningiomas might be considered circumscribed and easily distinguishable lesions in which fluorescence might not play a meaningful role during surgery. However, considering tumors that may have bony infiltration, tumors infiltrating extracranial tissues, satellite lesions, recurrent tumors with scar tissue, those with a dural tail, or tumors infiltrating the arachnoid, additional reliable tissue information based on fluorescence might be beneficial. Such benefit may be derived from identifying residual tumor overlooked during conventional surgery. In meningiomas, progression-free survival has been linked to the degree of resection [11], and any additional detection of unexpected tumor will serve to increase the degree of resection.

Several investigators have described the use of 5-ALA for resection of meningiomas, citing a high diagnostic accuracy for fluorescing tissue to predict tumor and the propensity for using this fluorescence to help identify residual tumor otherwise missed [7,8,9,10]. However, all reports have been based on monocentric evaluations and unintentional bias cannot be ruled out. Specifically, bias might be involved in sampling strategies for determining diagnostic accuracy (sensitivity, specificity), or while identifying residual tumor. To this end, we have recently published possible pitfalls and biases which need to be observed for the demonstration of correlation between fluorescent agent and tumor tissue, especially in infiltrative lesions [12].

Regulatory approval will therefore depend on prospective, well-controlled studies with a minimum of bias, showing fluorescence to be closely associated with the presence of tumor cells and imaging agents to be toxicologically safe. A consensus report by the International Society of Image Guided Surgery recently compiled several recommendations for clinical trial designs regarding intra-operative imaging agents [13]. These recommendations reflect that the toxicity of such agents should be acceptable and at a level below that of a therapeutic agent but not as low as required for a purely diagnostic agent. Fluorescence should be strongly correlated with the presence of tumor, as assessed by standard detection methodology. Furthermore, there is agreement that for the agent to advance to routine clinical use, methods for detection should be widely adopted and standardized. The use of such imaging agents will also have to benefit patients, i.e., such agents will have to demonstrate clinical usefulness [13,14].

Widely used for fluorescence-guided resection of high-grade gliomas, 5-ALA is a well-characterized agent with low toxicity. Devices for visualizing fluorescence are similarly widely distributed, including neurosurgical microscopes adapted for fluorescence (e.g., Blue400, Zeiss Meditech, Oberkochen, Germany; FL400, Leica Microsystems, Heerbrugg, Switzerland) and a validated surgical loupe and illumination device [15]. Given past observations of specificity and sensitivity in meningiomas, 5-ALA appears to be an agent that qualifies for a larger-scale study in patients, not only for a prospective demonstration of diagnostic accuracy and safety, but also for providing an indication of clinical benefit.

Here we present a clinical study designed to provide evidence of the usefulness of 5-ALA-induced tumor fluorescence for meningioma resection, looking at two aspects for a demonstration of benefit, i.e.,:

A: Determining the incidence of unexpected residual fluorescing tumor not distinguished under conventional white light;

B: Determining instances in which fluorescence better predicts tissue type (tumor or not tumor) in tissues with uncertain tissue origins, as assessed by conventional white light by the surgeon, e.g., scar versus tumor, hyperemia versus infiltration in the dural tail, or reactive hyperostosis versus bony tumor infiltration. Better prediction of this sort could change surgical strategy, not only if a tissue fluoresces, signifying tumor, but also if tissue fluorescence is missing, signifying normal tissue, which will then not necessarily be resected.

To reduce bias, this study employs a peer review panel, which anonymously reviews all surgical assessments and decisions, and the assessments of which provide the basis for the final study results.

## 2. Materials and Methods

### 2.1. Study Design

The NXDC-MEN-301 phase 3 open-label single-arm study is designed to investigate the safety, diagnostic performance, and clinical usefulness of the imaging agent Gleolan™ (Aminolevulinic Acid Hydrochloride, ALA HCl, ALA, 5-ALA), an orally administered imaging agent for the real-time detection and visualization of meningiomas during tumor resection surgery. All patients with imaging suggesting meningioma, irrespective of grade or tumor location, are eligible. Key inclusion criteria include age >18 years and normal organ function, with a focus on liver and normal bone marrow function. Key exclusion criteria are hypersensitivity to porphyrins, the use of photosensitizing drugs (St. John’s wort, griseofulvin, thiazide diuretics, sulfonylureas, phenothiazines, sulphonamides, quinolones, and tetracyclines), or planned embolization of tumors prior to resection. Study surgeons are selected based on experience of 5-ALA fluorescence-guided resection and being equipped with a standard microscope for fluorescence-guided resection (Blue400, Zeiss Meditech, Oberkochen, Germany; FL400, Leica Microsystems, Heerbrugg, Switzerland).

Patients scheduled to undergo meningioma resection receive an oral solution of Gleolan^TM^ (5-ALA; 20 mg/kg bodyweight) 3 h (target range 2.5 to 3.5 h, maximally 5 h) prior to anesthesia and then undergo surgery. The exact time point of administration is recorded.

During the operation, surgeons are required to follow a highly defined sequence of activities for finding fluorescing tissue and during tissue collection as a basis for assessing clinical usefulness and the diagnostic accuracy of fluorescence. These activities are recorded by the integrated high-definition video cameras of either microscope (Blue400, Zeiss Meditech, Oberkochen, Germany; FL400, Leica Microsystems, Heerbrugg, Switzerland) for later peer panel review.

All time points of assessments and biopsies are documented. Prior to activation of a site, study surgeons at that site are closely educated regarding sampling algorithms but also minimizing tissue exposure under white light to reduce potential photobleaching of tissue porphyrins.

All biopsies are forwarded for blinded tissue analysis, including Ki67 proliferation indices and other molecular investigations, to the Institute for Neuropathology, University of Münster.

### 2.2. Observation and Biopsy Sequences


*Assessment of bulk tumor fluorescence with correlative pathology:*


The aim of fluorescence assessment and tissue biopsy in bulk tumor is to determine whether a tumor shows fluorescence and to correlate fluorescence with tumor histology. To collect this biopsy, surgeons are asked to switch to blue light as soon as bulk tumor is encountered during surgery, to determine whether any fluorescence is visible and to take a biopsy from a fluorescing part of the tumor. A neuronavigation screenshot is taken from the exact location of the biopsy (Figure 1). The entire sequence is recorded using the microscope’s video system.

2.
*Interrogation of “indeterminate” tissues:*


During the course of conventional white light surgery, the surgeon might encounter tissue which appears uncertain. From such “indeterminate” tissue the surgeon might require more information based on fluorescence. In such cases, the observation of fluorescence might lead to a change in surgical decision or surgical strategy. Such instances of indeterminate tissue may be found at the dural tail, in scar tissue with cases of recurrent meningioma, in the bone flap or at bony resection margins. In cases in which the surgeon requires more information based on fluorescence, the region in question is marked by the navigation pointer and a screenshot is taken. At this stage, the surgeon records his presumed diagnosis for the tissue area he or she desires more information for (“likely” or “unlikely” tumor) and his surgical decision based on his impression under white light (to resect or not to resect). The surgeon then switches to blue light, records whether he or she sees fluorescence, and states whether he or she plans to change surgical strategy based on fluorescence status. Under white light, the uncertain tissue is then biopsied for later histological evaluation. The entire process is recorded using the microscope’s video camera for later assessment by the review panel (Figure 2).

3.
*Identification of residual tumor at the end of surgery:*


After clearing a field from tumor under white light, with the surgeon no longer being able to identify residual tumor in that field of view, the surgeon has the possibility of switching to blue light to determine whether any pathological tissue can be identified based on fluorescence. If the surgeon finds residual fluorescing tissue after switching to blue light, this area is pointed out using the neuronavigation pointer (and a navigation screenshot is taken). A biopsy is taken from the exact same region after switching back to white light if this can be performed safely. This procedure is also recorded by video from beginning to end. Such interrogations might be performed at multiple stages of surgery, provided a particular area (field of view) has been cleared of tumor according to the impression of the surgeon under white light. The interrogation of an inconspicuous bone flap also constitutes an “end of surgery” scenario if the surgeon wants to rule out any tumor infiltration in the bone. If any fluorescent bone is encountered, then this is biopsied prior to drilling of the bone flap (Figure 3).

### 2.3. Bias Minimization

#### 2.3.1. Sources of Bias

Study surgeons collect tissue and make initial assessments regarding tissue characteristics (likely tumor or unlikely tumor) and fluorescence status, and these assessments are all documented in detail. However, unintentional bias cannot be completely ruled out in the unsupervised setting of individual surgeons in the operating room.

For instance, while surgeons are interrogating one area with uncertain tissue characteristics (indeterminate tissue) they might encounter a different area in the same field of view with fluorescence which immediately leads them to think that the second area might show tumor. Once fluorescence has been observed, it cannot simply be unseen, and this will influence any further decisions.

Secondly, fluorescence is perceived by the surgeon but it is not objectively quantified. Therefore, a study requires mechanisms for ensuring that any fluorescence or lack thereof in the surgeon’s perception is objectively verified.

An important aspect of the study is calling out areas of tissue where the surgeon is uncertain about tissue type (likely or unlikely tumor) as “indeterminate” tissue. A non-compliant surgeon might simply call out an area of bulk tumor as being “indeterminate” in the unsupervised setting to expedite his study activities. If such tissue then also shows fluorescence, this will favor endpoints regarding diagnostic accuracy and the capability of 5-ALA-induced fluorescence for identifying tissue better than the surgeon using conventional light. The same might be true for any clearly normal tissue areas unrelated to tumor and without fluorescence.

While interrogating regions of the surgical cavity for residual tumor based on fluorescence (end-of-surgery assessment), the surgeon must initially state that he or she sees no residual tumor under white light. This statement must formally be considered as subjective and therefore requires confirmation by an independent reviewer. A surgeon might unintentionally overlook obvious residual tumor which would then fluoresce and be biopsied, again skewing the study data in favor of 5-ALA.

Finally, if a surgeon interrogates a specific area for fluorescence and these findings are recorded, the surgeon is required to take a biopsy from the questionable area. If he or she takes the biopsy from a slightly different area (inadvertently), then obviously such a biopsy would not be a valid representation of the underlying tissue diagnosis or the surgeon’s assessment of tissue type (tumor likely or unlikely).

#### 2.3.2. Methods for Bias Reduction

As in many studies, all histological examinations in this study are rater blinded, i.e., the central study neuropathologist is unaware of declarations regarding assumed tissue type by surgeons and fluorescence data. The histology of biopsy samples (meningioma or normal tissue) is used as the “truth standard” for the determination of diagnostic performance and diagnostic accuracy of 5-ALA-induced PpIX fluorescence. Blinding of pathologists, however, will not overcome biases intentionally or non-intentionally introduced by the unsupervised surgeon.

Therefore, this study, for the first time in the context of a study for intra-operative fluorescence imaging, employs a completely independent review panel (“biopsy review panel”) of peer neurosurgeons who are unaware of each other’s identities and who are unfamiliar with any of the study surgeons or the study’s principal investigators. The reviewers, whose role it is to make their own assessments of tissues and fluorescence status, are also blinded to any of the surgeons’ intraoperative assessments. Importantly, data for the pivotal analyses will be derived from the assessments of the reviewers.

Study reviewers are supplied with various segments of the videos acquired during surgery in a random order, with video segments from a single case not being reviewed en bloc by a single reviewer but rather being split and distributed among reviewers. Regarding additional data, reviewers are acquainted with the patient’s age, whether the patient has a recurrent or de novo tumor, the orientation of the patient in space, and, for each segment, the neuronavigation screenshot correlating with the video segment or any biopsies.

#### 2.3.3. Tasks of the Biopsy Review Panel

For tissues considered “indeterminate” by study surgeons, reviewers are independently asked whether they consider the tissue area as being likely or unlikely to represent tumor. The same holds for assessing areas cleared of tumor under white light, in which case reviewers are asked if they see any residual tumor in the respective video segments.

For video segments obtained under blue light, in which an indeterminate area is being pointed out by the neuronavigation pointer, the reviewers are asked to state whether they see any fluorescence or not. Similarly, in video segments documenting the interrogation of an end-of-surgery field of view (considered cleared from tumor under white light), the reviewers are asked if they see fluorescence under blue light. Assessment of fluorescence is also carried out for mandatory biopsies taken from the bulk tumor.

Finally, if surgeons identify specific areas during their interrogation of “indeterminate” tissues or fluorescing tissues at the end of surgery, it is the reviewers’ role to independently verify that biopsies are taken from the respective area of interest. For this, reviewers have access to all necessary video and image material.

For quality control, raters are periodically presented with mock data sets with distinct contents for determining the reproducibility of raters’ answers. Raters will be naïve to when such mock data sets are intermingled with study data sets.

#### 2.3.4. Elimination of Possibly Biased Biopsies from the Biopsy Efficacy Analysis Population

Biopsies are eliminated on account of possible bias when the location of the tissue that is biopsied does not correspond to the fluorescing tissues at the end of surgery or the location of a biopsy does not correspond to the tissues pointed out as being “indeterminate” by study surgeons before assessing fluorescence status.

When interrogating “indeterminate” tissue regions, biopsies will be eliminated from the Biopsy Efficacy Analysis Population if all independent reviewers, together with the study surgeon, are unanimous in their judgement regarding tissue type (likely or unlikely tumor) and the fluorescence finding corresponds to final tissue histology. Presuming that the opinions of reviewers will be divergent in truly “indeterminate” regions, under such circumstances the biopsy is eliminated for possible obviousness. A theoretical example of this would be if the surgeon incorrectly interrogates bulk tumor as “indeterminate”, the surgeon and all blinded reviewers consider the area to be tumor, the tissue fluoresces, and the pathologist diagnoses tumor.

In a similar setting, on the other hand, if a single assessment diverges from the others, the biopsy will be valid and will be used for calculating the endpoints of the study. An example of this would be when the study surgeon claims that a tissue is “indeterminate” and judges the tissue to be likely to represent tumor, all reviewers come to the same conclusion (“likely tumor”), the location fluoresces, but the pathologist fails to find tumor. Eliminating this biopsy would favor 5-ALA, since a tissue sample with false-positive fluorescence would be eliminated.

### 2.4. Study Aims and Biometry

The primary study aim is to determine the percentage of participants for which 5-ALA-induced PpIX fluorescence status allows the surgeon to visually obtain correct information as to the presence or absence of tumor in tissue where there is uncertainty regarding that tissue’s tumor status based on white light visualization alone.

We assume that if a minimum of 30% of study participants achieve the primary efficacy endpoint, this proportion is clinically meaningful. This means that at least 30% of the study participants will have at least one indeterminate or unexpected fluorescent end-of-surgery tissue area where 5-ALA-induced PpIX fluorescence status is consistent with histological assessment of meningioma. With 100 participants included in the per-protocol population, the lower bound of a 95% confidence interval of the success rate will be 40.4% using a Wilson (score) confidence interval. This lower bound is considered clinically meaningful in this population.

Key secondary endpoints are:The biopsy-level positive predictive value (PPV) of 5-ALA for the real-time visualization of tissue locations on the tumor margin in newly diagnosed or recurrent meningioma during resection surgery.To determine the biopsy-level negative predictive value (NPV) of 5-ALA for the real-time visualization of tissue locations on the tumor margin in newly diagnosed or recurrent meningioma during resection surgery.To determine the participant-level PPV of 5-ALA for the real-time visualization of bulk tumor in newly diagnosed or recurrent meningioma during resection surgery.To determine the biopsy-level diagnostic accuracy of meningioma identification with (i) 5-ALA-induced PpIX fluorescence status under BL vs. (ii) visualization under WL in indeterminate tissue and unexpected fluorescent EOS tissue locations, as assessed by the operating surgeon.

In this study, safety is assessed in a safety analysis population of all patients meeting the eligibility criteria for the study and who receive 5-ALA. Efficacy is assessed in a per-protocol population defined as having participants dosed with 5-ALA, who undergo surgery, have a histologically confirmed meningioma (all grades), and have at least one tissue biopsy included in the biopsy efficacy analysis population. For sensitivity purposes, all assessments at a biopsy level will also be performed in an all-biopsy population which includes any biopsies collected during surgery. The biopsy efficacy analysis population will only contain biopsies not eliminated due to possible bias.

Patients are followed for a total of 6 weeks to assess pharmacological safety of 5-ALA.

## 3. Results

In all, 16 centers are involved in this study, 14 from the United States, 1 from Germany, and 1 from Vienna. So far, 88 patients have been included in the trial.

No safety concerns have been observed up to now, and the average number of biopsies has exceeded the minimal threshold projected for this study.

Due to unforeseen technical failure of recording equipment and the fact that the primary study aims depend on the assessments of review panelists, who in turn rely on the video footage, approximately 10 patients will be recruited in addition to the originally projected 100 patients. Patients with whom technical failure of recording equipment is experienced will be removed from the primary analysis.

## 4. Discussion

Fluorescence-guided resection is a new field, first introduced in surgery for malignant gliomas and now in widespread use. In the context of malignant gliomas, visible residual 5-ALA-induced fluorescence identifies residual tumor, which may lead to more complete resections, as demonstrated in a prospectively randomized phase 3 study [3]. Surgical microscopes adapted for detecting fluorescence are widely available (Zeiss Meditech Blue400, Leica FL400). The range of available devices has recently been expanded by a loupe-based system (DVI Reveal), which has been validated as achieving the same diagnostic performance as the established microscopes [15]. Given regulatory approval for 5-ALA by the European Medicines Agency in 2007, followed by FDA approval in 2017 [16], 5-ALA, which is marketed as Gliolan in the EU and Gleolan^TM^ in the US, there has been considerable interest in extending FGR to other brain tumor types. Among these, based on the literature, meningiomas may be a promising candidate [5,6,7,8,9,10].

Apart from brain tumors, efforts are underway to extend FGR to other organ systems as well, not only using 5-ALA [17,18,19,20], but also other fluorochromes, and study designs resulting in regulatory approval are being intensely discussed [21].

Any newly introduced agents or applications of available agents for intraoperative tumor imaging will have to be tested for toxicological safety and will require demonstrations of diagnostic accuracy and clinical usefulness. Such usefulness will be assumed when the agent helps to identify residual tumor in diseases where the extent of resection is accepted to be linked to patient outcomes but will not have to be linked to changes in outcome, i.e., longer survival or progression-free survival. On the other hand, simply showing that tumor is visualized by a fluorochrome, for example, will not be sufficient [14].

Prior to the present study, we have discussed several study designs for demonstrating the clinical usefulness of 5-ALA FGR in the treatment of meningiomas. Theoretically, usefulness could be linked to improvements in overall survival or progression-free survival as endpoints for a randomized study. In the context of meningioma, progression-free survival (PFS) is beyond 10 years [22], depending on degree of resection and tumor grade, so that PFS as endpoint would not allow a study to be completed in a meaningful period of time. A different approach may be a linkage of study endpoints to the completeness of resection with early post-operative MRI. However, even from very early studies [23], it is well accepted that there are differences in PFS that are related to whether the bulk tumor is removed with underlying dura and bone (Simpson Grade I), the dura is merely coagulated (Simpson II), or the tumor is removed without the dura being coagulated (Simpson III). It would be exceedingly difficult to objectively differentiate between Simpson Grades I to III resections on the basis of early post-operative MRI. Furthermore, in many complex meningioma cases, tumors are not completely removed if neurovascular structures are endangered, without cytoreductive surgery being viewed as being futile.

Designing a study to demonstrate diagnostic accuracy in an unbiased fashion is highly challenging since surgeons cannot be “blinded” to an intra-operative imaging method. Even the most complex design for such a study that can be envisaged is not free from possible biases. Such a study could be randomized and placebo-controlled, with surgeons being blinded to study group allocation. Surgeons would first operate under white light. After declaring the end of resection under white light, surgeons would change to the fluorescence mode for detecting residual fluorescence, which is subsequentially addressed by resection of or biopsies from fluorescing tissue. A possible endpoint for this study would be the incidence of residual tumor detected under blue light. This study design stipulates that surgeons will try to resect equally thoroughly in both study arms, being unaware of final study group allocation.

The unsupervised surgeon could just operate less thoroughly under WL to ensure that there is residual tumor to be detected. It would be difficult to keep surgeons from switching to blue light before completing white light resection, and observation of any fluorescence would immediately bias the surgeon with respect to the location of possible residual tumor. Furthermore, this study design would miss any additional benefit from switching anytime for tissue interrogation during the course of surgery, which might change surgical strategy. There is an advantage to correctly identifying healthy tissue from its lack of fluorescence, e.g., in scar tissue, and thus correctly deciding against removal. Photobleaching of fluorochromes during surgery under white light or destruction of fluorochromes by coagulation, drilling, or use of an ultrasound aspirator, by destroying fluorescence, may lead to false-negative results.

Due to the difficulties in effectively blinding surgeons during studies on fluorescence-guided resections to reduce the risk of introducing inadvertent or deliberate bias, the operating surgeons are not dependable as raters and any data generated by the surgeons may not be acceptable as objective. Thus, we stipulated that the surgeons have to be replaced by external raters who have access to intra-operative imaging such that bias is minimized. A rater panel will, in an independent fashion, assess whether tissue that a surgeon cannot confidently interpret under white light (i.e., “indeterminate” tissue) is indeterminate in the panel’s eyes, if any fluorescence observed by the surgeon is also reproducibly observed by the raters and biopsies are collected from exactly those regions pointed out as being indeterminate or showing unexpected end-of-surgery fluorescence.

### Limitations

The present study design has several confounders and limitations. First of all, the study considers the standard 5-ALA dose of 20 mg/kg. Although toxicological safety has been demonstrated for higher doses of 5-ALA [24], it is unknown whether higher doses of 5-ALA would provide more fluorescence in meningioma. The present study will not allow the generation of data to this end, which might be the subject of follow-up trials.

The study relies on study surgeons actively interrogating areas in which they would consider additional information from fluorescence to be helpful. This study does not force surgeons to perform biopsies, except for biopsies from bulk tumor, because mandating biopsies from any intracranial tissues would be unethical. Theoretically, a surgeon could perform the entire procedure without interrogating tissue once, except for the bulk tumor. This inherent uncertainty regarding the number of biopsies and, respectively, the productive participation of surgeons can only be resolved by the careful education of surgeons about study procedures and study aims.

The study is designed as an all-comer study, irrespective of meningioma grade, anatomical location, size, or urgency of surgery. Since the obligatory assessment of each tissue interrogation/biopsy is reviewed by external panelists and their assessments are based on videos, video quality must be high. For deep-seated tumors, such as petroclival meningiomas, which are operated under high magnifications, optimal conditions for obtaining evaluable videos, specifically under blue light, are difficult to achieve. Retraction for the sake of better video imaging would not be considered acceptable for obvious reasons. It is likely that if patients with such deep-seated tumors are recruited, the number of biopsies qualifying for evaluation will be low even if there is useful fluorescence.

Furthermore, inclusion in this study is based on pre-operative MR imaging of a meningioma. With minor exceptions, the pre-operative assumption will be correct. However, in rare cases, other pathologies might mimic meningioma and result in erroneous recruitment [25,26], such as carcinoma metastasis, dura lased malignant gliomas, lymphomas, plasmocytomas, undifferentiated sarcomas, chondrosarcoma, or schwannomas. Such patients would be included in the safety population but would obviously not qualify for the prespecified endpoints. Regarding the different grades of meningioma, it is difficult to pre-operatively reliably predict histological differentiation. Past experience has suggested that fluorescence accumulates independently of meningioma grade [9]. Due to the relative paucity of high-grade meningiomas compared to grade I meningiomas, the number of malignant subtypes in this study will be small.

In some centers, meningiomas are embolized preoperatively in order to reduce blood loss and to reduce the complexity of surgery [27]. We hypothesize that due to reduced blood flow to tumors or necrosis of tumor tissue, drug delivery to the tumor would be impaired. Thus, such patients are excluded from this study, and it will be unknown whether any benefit from FGR extends to embolized tumors.

Finally, this study relies on perfectly functioning video documentation. We have encountered operations in which the microscope videos did not work sufficiently. Under these circumstances, panel review cannot occur. Data from such patients will be included in the safety population and in separate exploratory evaluations of data based on the surgeons’ impressions only but will be excluded from analyses based on the Biopsy Efficacy Analysis Population. To compensate for these patients, the number of patients enrolled in the study has been slightly expanded.

## 5. Conclusions

In the field of intra-operative imaging, a number of biases and confounders complicate the generation of objective data for proving diagnostic accuracy and clinical benefit. This study provides a novel approach for demonstrating clinical benefit with an algorithm for independent review to overcome concerns about critical influence being exerted by unsupervised surgeons. Patient recruitment, enrollment, and processing history to date confirm that the study design presented is feasible. We anticipate that the last patient will be enrolled in Q4 2022.

## Figures and Tables

**Figure 1 brainsci-12-01044-f001:**
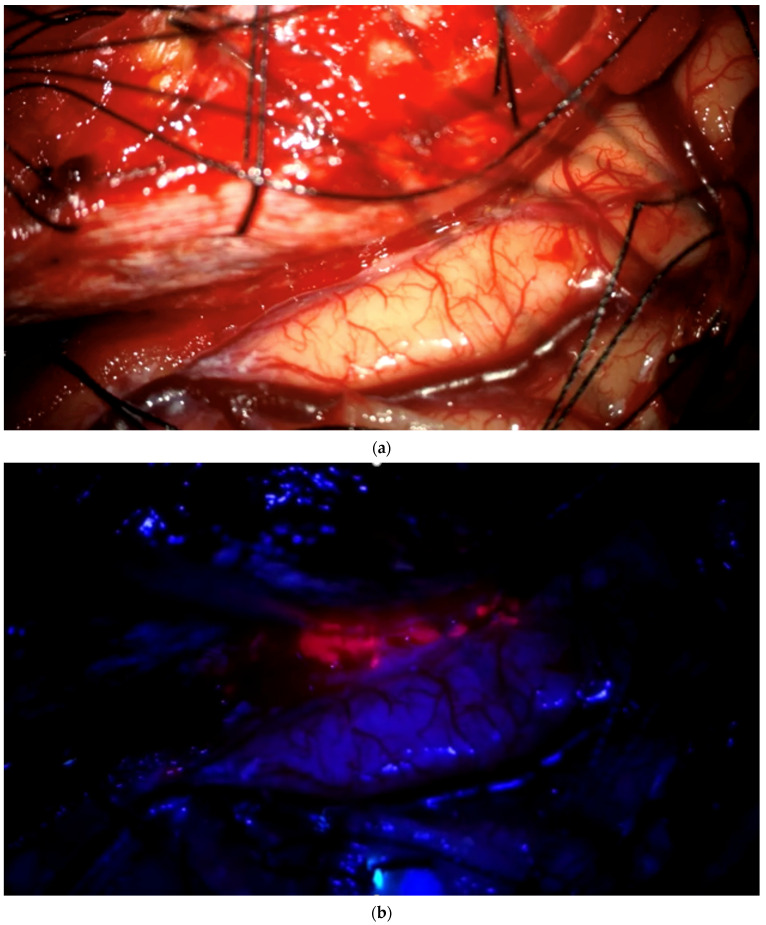
Biopsy acquisition from bulk tumor. As soon as surgeons encounter bulk tumor during surgery, they change to blue light, determine whether any fluorescence is visible, and, if so, take a biopsy from fluorescing tissue after taking a neuronavigation screenshot. The biopsy procedure is recorded for later assessment by the review panel. (**a**) Brain with visible bulk meningioma under elevated dura in a patient with tumor in the superior sagittal sinus infiltrating the adjacent brain. Insert shows pointer tip location marking the intended biopsy location. (**b**) Strong fluorescence of tumor under blue light, which is biopsied. (**c**) Neuronavigation screenshot (“F”: footwards; “H”: headward; “A”: anterior; “P”: posterior; “L”: left).

**Figure 2 brainsci-12-01044-f002:**
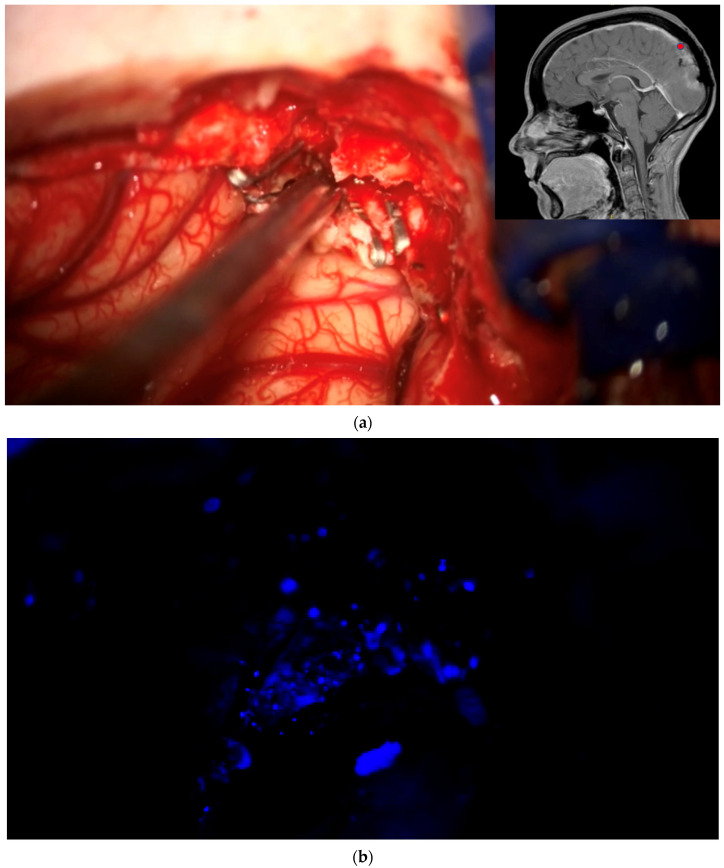
Biopsy from “indeterminate” tissue: In this study the surgeon might encounter tissue which appears uncertain (tumor or not, “indeterminate” tissue) during surgery and where fluorescence might be helpful. The location of a respective region is documented by the neuronavigation screenshot. The illumination is changed to blue light to determine whether the tissue fluoresces and a biopsy is then taken. The tissue interrogation procedure is recorded by video. (**a**) In this example a portion of the sagittal sinus infiltrated by meningioma has been resected. The anterior resection margin appears inconclusive as to whether it still contains tumor and is therefore regarded as being indeterminate. The location is documented using neuronavigation. The insert shows the pointer tip location marking the intended biopsy location. (**b**) No fluorescence is observed. The location is then biopsied. (**c**) Neuronavigation screenshot “A”: anterior; “P”: posterior; “L”: left; “R”: right).

**Figure 3 brainsci-12-01044-f003:**
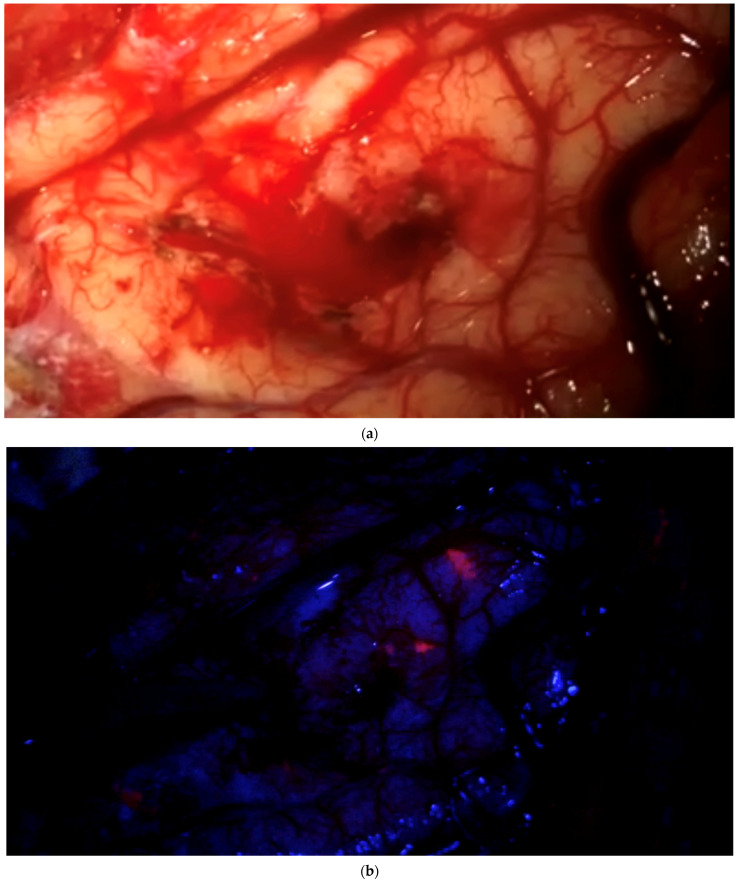
“End of surgery” biopsy: After clearing a field from tumor under white light, with the surgeon no longer being able to identify residual tumor in that field of view, the surgeon switches to blue light to determine whether any pathological tissue can be identified based on fluorescence. Any fluorescing tissue is pointed out using the neuronavigation pointer (and a navigation screenshot is taken). A biopsy is then taken collected. This procedure is recorded by video from beginning to end. (**a**) After dissecting tumor from the brain, the brain is considered free of tumor. (**b**) Various areas of fluorescence can be observed. The surgeon would then resect the fluorescing tissue shown. (**c**) Navigation pointer and corresponding screenshot (“F”: footwards; “H”: headward; “A”: anterior; “P”: posterior; “L”: left; “R”: right).

## Data Availability

Not applicable.

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
