# Peer review of "The NXDC-MEN-301 Study on 5-ALA for Meningiomas Surgery: An Innovative Study Design for the Assessing the Benefit of Intra-Operative Fluorescence Imaging"

_brainsci, 2022, doi:10.3390/brainsci12081044_

Round 1

Reviewer 1 Report

The manuscript presents the study protocol of a Phase 3 open-label single-arm study investigating the feasibility and impact of 5-ALA FGR in intracranial meningiomas. As the manuscript hereby reviewed does not show any preliminary result of the investigation, no critical review can be made on the usefulness of the study. 

The protocol is properly designed and described. The rationale of the study and the literature introduction question the applicability of a well-recognised surgical resource (in glioma surgery) to a novel tumour type to enhance surgeons' perspective of bulk and residual tumor at the beginning and final inspection during surgery. 

The methods are correct, meticulously described and critically generalised according to previous literature, study design limitations and near-unavoidable biases. As correctly stated by the authors, the main issue of such a protocol is limiting as much as possible the Hawthorne effect of study surgeons when completion of resection or indeterminate tissue encounters are to be declared. Despite proper ethical and methodological adherence of physicians to the protocol, related biases must be addressed during study design.  I personally appreciated the employment of an indirect review panel to check the classification of tissues visualizations under both standard and blue lights. 

Given my previous comments, I judged the manuscript to be of clear clinical interest and eligible for publication. 

Corrections:

Line 384: Such usefulness will be assumed 383 when the agent helps to find identify residual tumor. The phrase results unclear, please correct. 

Author Response

Reviewer  1

Line 384: Such usefulness will be assumed 383 when the agent helps to find identify residual tumor. The phrase results unclear, please correct. 

-This sentence has been corrected. We thank the reviewer for his comments.-

Reviewer 2 Report

Medical English use is excellent and exposition is logical and orderly. Several minor suggestions and comments below.

Fig. 1 is beautifully done. 

Re. Fig. 3 b, would the authors add in legend the comment that “the surgeon would then resect the shown fluorescing tissue” ?

Line 39, don’t abbreviate “ALA”. Use “5-ALA” always. 

Line 229, I am unclear what the authors mean by “must be independently confirmed”.

Line 230 might be better stated “unintentionally”.

Line 393, PFS was not previously defined. Although any clinician reading this will know this abbreviation, it should probably still be spelled out prior to first use to help beginners and non-clinical readers.

Line 425, CUSA must be spelled out, cavitron ultrasonic surgical aspirator. Most neurosurgeons will know this but even medical neurooncologists might not.

Line 426, the fault could be mine, but I didn’t understand the sentence

“The potential advantage of also correctly identifying non-involved tissue by lack of fluorescence (and thus not leading to removal) cannot estimated.“.  Could the authors rephrase ?

The problem of photobleaching is an abiding problem in FGR. Even the operative white light contributes to this. I would mention this in line 425 also, if the authors agree with my estimation. To potentially address the problem of photobleaching, would it help to document total time exposed to operative white light when every false negative [Bx pos., fluorescence neg] was taken ? If the time interval average of these was significantly longer than the Bx pos., fluorescence pos. samples, this might suggest photobleaching ? We could then institute light intensity reductions or filters to reduce the bleaching ?

Re. the proposed use of 5-ALA 20 mg/kg, I would ask the authors to consider comparing 20 mg/kg with 50 mg/kg. The argument for using a higher dose is sound, with potential for allowing more complete resections. My personal estimation is that risks of this higher dose are within International Society of Image Guided Surgery guidelines. See references 1 and 2 below.

Considering the preclinical data of Fe chelation increasing 5-ALA mediated fluorescence, would the authors think it worthwhile amending their study to include deferiprone [the one marketed brain penetrant Fe chelator] several days prior to surgery to determine if this would increase fluorescence and completeness of resection ? 

Ideally the authors should mention that they will compare tissue fluorescence with Ki67.

I recommend that the exact time of surgeons’ evaluation and of oral 5-ALA intake be recorded and that the authors look for any correlation between time interval and fluorescence. We cannot assume that time-to-peak fluorescence in meningiomas is the same as for GBs.

_____________________________

1: Michael AP, Watson VL, Ryan D, Delfino KR, Bekker SV, Cozzens JW. Effects of 5-ALA dose on resection of glioblastoma. J Neurooncol. 2019 Feb;141(3):523-531. doi: 10.1007/s11060-019-03100-7.

2: Cozzens JW, Lokaitis BC, Moore BE, Amin DV, Espinosa JA, MacGregor M, Michael AP, Jones BA. A Phase 1 Dose-Escalation Study of Oral 5-Aminolevulinic Acid in Adult Patients Undergoing Resection of a Newly Diagnosed or Recurrent High-Grade Glioma. Neurosurgery. 2017 Jul 1;81(1):46-55. doi: 10.1093/neuros/nyw182. 

Author Response

Reviewer 2

Medical English use is excellent and exposition is logical and orderly. Several minor suggestions and comments below.

Fig. 1 is beautifully done. 

Thank you, also for your close review and highly appreciated comments and suggestions.

Re. Fig. 3 b, would the authors add in legend the comment that “the surgeon would then resect the shown fluorescing tissue” ?

The legend was amended

Line 39, don’t abbreviate “ALA”. Use “5-ALA” always. 

This has been fixed, along with another instance further down in the text

Line 229, I am unclear what the authors mean by “must be independently confirmed”.

I have rephrased and expanded the sentence:

"While interrogating regions of the surgical cavity for residual tumor based on fluorescence (end of surgery assessment), the surgeon must initially state that he or she sees no residual tumor under white light. This statement must formally be considered as subjective and therefore requires confirmation by an independent reviewer. A surgeon might non-intentionally overlook obvious residual tumor which would then fluoresce and be biopsied, again skewing the study data in favor of 5-ALA."

Line 230 might be better stated “unintentionally”.

Corrected

Line 393, PFS was not previously defined. Although any clinician reading this will know this abbreviation, it should probably still be spelled out prior to first use to help beginners and non-clinical readers.

Corrected

Line 425, CUSA must be spelled out, cavitron ultrasonic surgical aspirator. Most neurosurgeons will know this but even medical neurooncologists might not.

Has been changed to "ultrasound aspirator"

Line 426, the fault could be mine, but I didn’t understand the sentence

“The potential advantage of also correctly identifying non-involved tissue by lack of fluorescence (and thus not leading to removal) cannot estimated.“.  Could the authors rephrase ?

The paragraph now reads:

“There is an advantage to correctly identifying healthy tissue from its lack of fluorescence, e.g. in scar, and thus correctly deciding against removal. Photobleaching of fluorochrome during surgey under white light, or destruction of fluorochrome by coagulation, drilling, or by using the ultrasound aspirator during destroys fluorescence, possibly leading to false negative results.”

The problem of photobleaching is an abiding problem in FGR. Even the operative white light contributes to this. I would mention this in line 425 also, if the authors agree with my estimation.

We agree completely and have added this aspect to the paragraph (see above).

To potentially address the problem of photobleaching, would it help to document total time exposed to operative white light when every false negative [Bx pos., fluorescence neg] was taken ? If the time interval average of these was significantly longer than the Bx pos., fluorescence pos. samples, this might suggest photobleaching ? We could then institute light intensity reductions or filters to reduce the bleaching ?

This is an important point brought up by the reviewer. However, we cannot completely control for photobleaching at this point during the study and, in a phase III study, the general use of 5-ALA for FGR would have to be robust enough to withstand such influence.  We did, however, educate all surgeons on reducing light as far as possible to try to minimize bleaching, but cannot control for this and habe added the following statement:

Line 133: “All time points of assessments and biopsies are documented. Prior to activation of a site, study surgeons at that site are closely educated regarding sampling algorithms, but also on minimizing tissue exposure under white light to reduce potential photobleaching of tissue porphyrins.”

Re. the proposed use of 5-ALA 20 mg/kg, I would ask the authors to consider comparing 20 mg/kg with 50 mg/kg. The argument for using a higher dose is sound, with potential for allowing more complete resections. My personal estimation is that risks of this higher dose are within International Society of Image Guided Surgery guidelines. See references 1 and 2 below.

Considering the preclinical data of Fe chelation increasing 5-ALA mediated fluorescence, would the authors think it worthwhile amending their study to include deferiprone [the one marketed brain penetrant Fe chelator] several days prior to surgery to determine if this would increase fluorescence and completeness of resection ? 

Regarding higher doses, we have added a statement in the “Limitations” section of the manuscript as this is an interesting point and added reviewer's
Reference Nr. 2 to the manuscript. Unfortunately our study has already started and can no longer be amended in a simple fashion,  but it seems highly reasonable and interesting to consider higher doses or other medical interventions, such as Fe chelators in future studies.

“The present study design has several of confounders and limitations. First of all, the study considers the standard 5-ALA dose of 20 mg/kg. Toxicological safety has been demonstrated for higher doses of 5-ALA also [24] and it is unknown whether higher doses of 5-ALA would provide more fluorescence in meningioma. The present study will not allow generating data to this end, which might be the subject of follow-up trials.”

[24] Cozzens JW, Lokaitis BC, Moore BE, Amin DV, Espinosa JA, MacGregor M, Michael AP, Jones BA. A Phase 1 Dose-Escalation Study of Oral 5-Aminolevulinic Acid in Adult Patients Undergoing Resection of a Newly Diagnosed or Recurrent High-Grade Glioma. Neurosurgery. 2017 Jul 1;81(1):46-55. doi: 10.1093/neuros/nyw182. 

Ideally the authors should mention that they will compare tissue fluorescence with Ki67.

I recommend that the exact time of surgeons’ evaluation and of oral 5-ALA intake be recorded and that the authors look for any correlation between time interval and fluorescence. We cannot assume that time-to-peak fluorescence in meningiomas is the same as for GBs.

This is an important point and fortunately, these data are being recorded exactly. With have added the appropriate language to include this aspect:

Line 134 ff: “Also, the time points of all assessments and biopsies are documented. Prior to activation of a site, study surgeons at that site are closely educated regarding sampling algorithms, but also on minimizing tissue exposure under white light to reduce potential photobleaching of tissue porphyrins.

All biopsies are forwarded for blinded tissue analysis, including Ki67 proliferations indices and other molecular investigations, to the Institute for Neuropathology, University of Münster.  “

_____________________________

1: Michael AP, Watson VL, Ryan D, Delfino KR, Bekker SV, Cozzens JW. Effects of 5-ALA dose on resection of glioblastoma. J Neurooncol. 2019 Feb;141(3):523-531. doi: 10.1007/s11060-019-03100-7.

2: Cozzens JW, Lokaitis BC, Moore BE, Amin DV, Espinosa JA, MacGregor M, Michael AP, Jones BA. A Phase 1 Dose-Escalation Study of Oral 5-Aminolevulinic Acid in Adult Patients Undergoing Resection of a Newly Diagnosed or Recurrent High-Grade Glioma. Neurosurgery. 2017 Jul 1;81(1):46-55. doi: 10.1093/neuros/nyw182.